# Curatorial Ethics and Indeterminacy of Practice

**Sasha Burkhanova-Khabadze** [1,2]

1    Exposed Arts Projects, London W8 4LY, UK; hello@exposedartsprojects.com or
     s.burkhanova0520161@arts.ac.uk or sashaburkhanova@khabadze.com; Tel.: +44-778-754-4522
2    Central Saint Martins, Fine Art Department, University of the Arts London, London N1C 4AA, UK

**Abstract:** This article defines "curatorial ethics" as a notion that has to be configured and constantly revisited by an independent curator throughout her practice. By inquiring into the personal motives, biases and drives, she would establish her own ethical position, convert it into a professional ethic and apply it to judge her own professional performance and her colleagues. Such perspective opposes the traditional understanding a professional ethic as a set of unitary guidelines to be passed to specialists (i.e., via education or early career). The notion of curatorial responsibility is redefined accordingly, and with conceptual inspiration from Gilles Deleuze and Karen Barad's concepts of "becoming" (Deleuze) and "intra-action" (Barad). A curator is addressed as accountable for configuring her practice in response to agendas and actions of other parties involved in the art project. That is, for facilitating the co-constitution of individual subject positions and practices via opening up herself to the terrors and potentials of unprecedented self-transformation.

**Keywords:** ethics; indeterminacy; curating; Deleuze; Barad

---

## 1. "Curatorial Ethics" as a Material-Discursive Composition

*Uncertainty is knowledge. Determining the limits of human perspective is not the same as asserting that we do not know or cannot know them.*

—Eduardo Velásquez, 2006 [1] (p. 150).

*The curator, that is to say, must ask herself not, Am I being a good curator (am I wild enough, am I orthodox enough, have I said and done the right things)? But, What kind of person do I want to be? There are plenty of people who will answer the first question for her. Faced with the second question, there may be terrors but there are no experts.*

—Mark Hutchinson, 2007 [2].

My urge to consider "curatorial ethics" as a material-discursive composition comes as an outcome of both my systematic analysis of the current state of curatorial discourse and practice, and my own experience obtained through day-to-day practice as an unaffiliated art curator. Such a twofold position implies an embodied need for a critical reflection on the experiences and performance as a curator; a reflection that ultimately discloses the interconnection between the questions "what is a curatorial ethic?" and "what is a curator?", suggesting that the way the latter question is answered will automatically impact the way the former will be then approached. In the end, curators' answers to these questions—whatever these would be—will always already entail a certain ontological responsibility: a responsibility for those theories of ethics, which they put together, enact and reproduce throughout their work and agency *as curators*. The theories of ethics, which independent curators choose to call "theirs", would not merely describe or represent the reality of curatorial practice; they would configure

it in a particular way, defining what *can be* accredited as curating today and affecting what will be recognised as curatorial potential in the future. That is to say, the currently dominating theories of ethics would result in the formation of a particular curatorial subject: a curator who is defined by the expectations of what she[1] should be; adjusted by limitations of what she can do. Where these come from? This is the (ethical) task for each curator: to find out.

It goes without saying that in the contemporary arts field the role of a curator is changing rapidly, suggesting that the question of curatorial ethics should always be kept open and responsive to the ongoing transformation of a curator's role. In this way, "securing" the ongoing state of fluidity for the curatorial subject—acting against its crystallisation and stabilisation into a univocal image—would be among the main tasks of curatorial ethics. Yet, there is a problem in sustaining such openness: once the conventional practices are established (in the work of an individual curator or curatorial community at large), bringing in experimentation would take more time, require mental and emotional effort, and consequently prove less productive.

What does she mean when she calls herself (or is called) a curator? How different are the contents of such conceptual (self)identification from what she actually does when she curates? How is she to treat with respect (instead of merely "solving" towards unification) the usual disagreement between a curator's aspirations (what she wants to do), capabilities (what she can do), doings (what she actually does), and performance (how it looks to others)? How to treat the effects of the discursive definitions of "curatorial power" (that is often equated with control, omnipresence and autocracy) on the *actual powers* (that is, the capabilities and agency) of an independent curator? It is by posing these questions over and over again that the seemingly obvious border between a "curatorial" (relevant, professional, accountable) and a "non-curatorial" (personal, irrelevant, momentary) action, decision, or statement of a curator can be blurred. After all, instead of either subjecting herself to this border or fighting against it, a curator may choose to enact a third possible subject position towards it: that of interrogating—what does the way I place this very border reveal about myself? About *my* ethic?

The notion of curatorial ethics balances on the fence between personal and professional ethics[2]. As is the case with other occupations, a professional ethic traditionally replaces personal ethic when an individual acts in the business environment, in her professional capacity. In curating, however, it is the personal ethic that becomes professionalised, with her values converted into professional-curatorial values. How does this conversion proceed? There is naturally a disagreement and tension between the two facets, well-captured by Suzana Milevska as she introduced two concepts: "becoming *a* curator" and "becoming-curator" [3]. Milevska defined the former as "a pragmatic decision to [...] make a living out of one of the 'sexiest' professions available in the international art world (one that focuses on singling out emergent art concepts, art objects and artists who produce either those concepts or objects)" [3] (p. 70); that is, to become a professional curator. She then opposed it with the latter: the understanding of curating as establishing "one's own position in the world as a thinking subject" that "opens up a new route in understanding contemporary art and the way both curators and artists position themselves in the contemporary art world and the world in general" [3] (p. 70). Milevska recognised "becoming a curator" as a particular intellectual and cross-disciplinary activity that incorporates theory,

---

[1] Throughout this article I have chosen to use "she/her" as pronouns to address an individual curator. This does not imply that the suggested perspective on curatorial ethics is not applicable to other genders, but highlights that the ideas presented are grounded in—and determined by—a personal, first-hand experience and practice of a female contemporary art curator.

[2] It is a professional ethic of independent (institutionally unaffiliated, freelance) curators that is addressed in this article. Unlike institutional curators, whose role and responsibilities are defined in their contracts and who are usually "supplied" by a code of practice by an institution or employer, the practice of independent curators develops in the conditions of indeterminacy. Nothing is formally prescribed to them: to either take on board, or argue and act against it. It is therefore the absence of any kind of formalised professional ethics for independent curators (on the one hand) and the multiplicity of controversial ethical expectations simultaneously coming from artists, institutions and other parties they encounter and/or collaborate with (on the other hand) that results in an ethically complex position I am addressing here. I ongoingly experience such tension in my work as an independent curator and observe it in my colleagues. The desire to reflect on this lived experience and interrogate it in-depth inspired the research leading to this article.

puts different artistic practices in juxtaposition with other disciplines, and makes new connections amongst the previously dispersed elements and agents. In another context, Milevska reflected on "becoming-curator" in more abstract terms: as a complex engineering process that implies "controlling the traffic of different libidinal economies, energies and phantasms" that are at work in the relations between artists, institutions and audiences during the process of exhibition production, which becomes "a platform where all these libidinal energies meet" [4].

While Milevska separated and compared the two identities of a curator, I suggest that they are profoundly intertwined and inseparable in daily curatorial practice. The separation then takes place when an individual curatorial practice is being incorporated into the curatorial discourse: a discourse, which (like any other) tends to systematise and homogenise its contents. As observed by Paul O'Neill, "if we are to consider the nature of our recent forms of practice as curation, it is always-already attempting to become curatorial discourse": that is, to become "transparent, visible, and self-critical" [5]. As curators aim at visibility and recognition, only specific facets of curating make it into the discourse. It is therefore not surprising that the emergent "slight, insubstantial, and personalised responses by curators on the subject of their own rarefied practice" [5] happen to be lost in this transition. In fact, the very possibility of curatorial practice as knowledge that can be systematised, critically assessed and taught relies of such filtration. As a result, the contemporary curatorial discourse predominantly covers such aspects of curatorial practice as exhibition-making, selecting artworks, working with artists, dealing with institutions, and communicating with audiences. It neglects the emerging openings in curation, missing the singular (sporadic, momentary, intuitive) practices that cannot be classified, letting these die out and disappear.

By recognising the task of curatorial ethics to embrace the complexity of singular curatorial practices, we may find a way to incorporate what is currently lost in the discourse of contemporary art curating: namely, the personal (intuition-driven, infra-rational) responses of curators that underlie their seemingly pragmatic, rational professional choices. We will thus understand the term "curatorial ethics" not as a set of principles that are meant to regulate the social-instrumental sides of one's dealings with artists, institutions, audiences and fellow curators. Rather, we will consider it as addressing the entangled relationship amongst the various paths of life, lived experiences, systems of values, agendas and beliefs carried by each individual curator in the first place: prior to, and outside of which the very term "curator" does not make sense.

## 2. A Curator as Someone Else

As I have argued elsewhere [6], it would be impossible to formulate a unitary code of ethics for curators due to the inherent indeterminacy of curatorial practice as such. It is a practice that cannot be, strictly speaking, bound to a common job description, with the object of its expertise (the uncertain field of contemporary arts) not necessarily bound to a particular subject matter, medium, historical period, or a species of collaborators (who may not necessarily be "artists"). In this regard, it is emblematic for curating that there will always be several roles and subject positions that overlap, merge and co-mutate. Moreover, their number would constantly grow, since each new variation of an individual practice would at once reproduce the profession at large and create a difference in terms of individual methodologies: the scope and kinds of projects one realises as a curator. In this sense, for instance, while the majority of curators today would be in some way engaged in the process of exhibition-making (reproducing the accredited curatorial subject and role), their curatorial practice is at the same time likely to include many other near-curatorial and previously non-curatorial activities, thus resulting in new ethical dilemmas to deal with. Today it is common that, together with the exhibition concept and presentation, curators' tasks would include fundraising, coordination, mass media communication management, conducting research, teaching at universities, reaching to and initiating new audiences for their work, and shaping a certain public image and reputation. A curator will be involved in the areas of education and politics of culture, participate in the marketing of the city where the project takes place, and promoting the city as a cultural centre. Each of these activities would enact a different role

and a different constellation of ethical dilemmas that are likely to coincide within a single curatorial project—but are unlikely to repeat in the exact same combination in another project, or be experienced by another curator.

Today, calling oneself a curator, a person is expected to elaborate, for in itself the term "curator" does not say much: it is indeterminate and invites configuration. There is a long history leading to this perspective. Back in the 1970s, Harald Szeemann, a pioneer of independent curatorship, has described the curator's professional identity as combining the roles of "administrator, amateur, speechwriter, librarian, manager and bookkeeper, conservator, financier and diplomat" [7]. In a sense, Szeemann not only came up with a list of jobs that coincided with his role; he also introduced an effective route of self-identification—via self-differentiation—that was enjoined by curators ever since. The specifics of Szeemann's approach resided in his vision of a curator as a fluid, reactive subject position, rejecting a common definition of the profession ("a curator is x"), but instead seeking to map its ongoing transformation in the course of a curatorial project: via comparison, analogy, and the exposure of the relations between curating and other kinds of engagement ("a curator is a, b, . . . , y, z"). In this way, in response to the various functions and states enacted in different circumstances and times, a curator was addressed as a consultant and knowledge manager [8], organizer, editor and compiler [9], catalyst, generator and motivator [10], producer [11], author, editor and critic [12], iconoclast [13] amongst others. The most popular, and probably the most problematic, has always been addressing a curator as an artist [14,15]. In the expanded title of the book by Natasha Hoare and Coline Milliard, *The New Curator* is presented at once as a researcher, commissioner, keeper, interpreter, producer and collaborator [16]. As has been summarised by artist Sylvette Babin, even though this condition of contemporary art curating is often mercilessly criticised or played up to a fault, a curator today increasingly tends to endure in various forms: "from professional curator to cultural practitioner occasionally taking on the role, independent curator to institutional curator, curator-author to artist-curator "; even more, "the role seems to adapt to every kind of exhibition and artistic event, and of course, to various institutional settings" [17]. Furthermore, as observed by Michel Vandevelde such halo of uncertainty is often strategically exaggerated: it allows contemporary curators to broaden their professional horizons and be involved in a wider variety of projects *as curators*, defining on-the-go what that attribution stands for in each particular project's case [18] (pp. 75–85).

Such indeterminacy of a curatorial role has been recognised as a problem by some authors. For instance, Paul O'Neill observed that there is indeed an ongoing tradition of addressing curatorial subjectivisation as "becoming somebody else"; in fact, this is "how we now understand curating" [5]. Critical of this tradition, O'Neill suggested that it is necessary for curatorial discourse and practice today that a curator is recognised *as a curator*—that is, without mimicking the discourses of others in order to produce her own [5]. Quite differently, the perspective I am advocating here suggests that the said indeterminacy is approached not as a problem that requires solving (i.e., by unifying curating, as suggested by O'Neill); rather, it is to be recognised as the essence of acting and thinking *curatorially*—that is, always in connection and in response to the professional environment, trying on different hats, taking after the diversity of disciplines, voices and outlooks one encounters. Practically, acknowledging the indeterminacy and learning to embrace it would be the ultimate survival strategy available for a curator. It is a special mechanism that enables the sustainability of an independent curatorial career, allowing it to successfully adapt to the unpredictability and constant renewal of the professional field of contemporary arts. Importantly, the notion of "curator as someone else" implies that we need to pay attention to the particular way in which the initially indeterminate subject (a generic curator) happens to be further "determined" through the projects in which the curator chooses to engage in. Only by first assessing this process of determination can we then evaluate the formation of curatorial ethics.

### 3. The Indeterminacy of a Curatorial Subject

The current state of the notion of "curatorial ethics" is a paradoxical one. While there is no agreement in place for how to define it and what is to be recognised as best curatorial practices (ethically speaking), the number of accusations of unethical curating within the professional art-curatorial community is proliferating. Many curators who develop original methodologies are said to be "either unethical or curate unethically" [19] (p. 20), claimed to act as exploiters of artists [14], be driven by cynicism and opportunism [20] (p. 217), and exhibit favouritism in choosing their collaborators [21]. Notably, in such cases, the attribution "unethical" does not mean infringing the best professional practices—there is no such thing. While sometimes it is confused with illegal actions [22] and bad curating [23], it most often stands for upsetting behaviour: in the art world "people often say "unethical" when they mean "this really ticks me off"" [24]. I have been several times called "unethical curator" myself in the last few years. In 2014, I curated an exhibition for a Russian state museum and was called "unethical" by a local art critic for including "offensive contents" in a G-rated display, while neither the museum nor myself considered it inappropriate. In 2016, as a part of my PhD project, I collaborated with an artist who at the time was my research supervisor; this behaviour was recognised as "unethical" by the University Ethics Committee. From 2018 when I started Exposed Arts Projects (an art-research hub and exhibition venue in London) I have been repeatedly called "unethical curator" by artists and collaborators who do not share the values and non-judgemental policy of my organisation. It may appear that such accusations—based on hurt feelings, personal disagreements and arbitrary judgements made by individuals—have nothing to do with professional ethics. Yet, from a different angle, they capture the peculiar operation of curatorial ethics and expose its dissimilarity to the conventional functioning of professional ethics employed by other disciplines.

Unlike the professional ethic of, say, a doctor, a lawyer or an engineer that is composed as a formal code (or its alternative) and is presented to a young practitioner as a part of their specialist education [22] (p. 8), from the early days of her career a young curator is encouraged to develop an ethic for herself. Curatorial courses are organised in a way to encourage students to "understand that they really need to want to do what they're doing" and "not make compromises in that" if they really want to become curators [25]: to be able to identify, access and make sense of their drives and desire to curate. Ruth Noack summarised the intention of a curatorial course she led at the Summer Academy of Fine Arts in Salzburg as to "instill . . . a kind of curatorial ethics" [25]. In a similar way, Beatrice von Bismarck, the course leader at Cultures of the Curatorial programme (Academy of Fine Arts, Leipzig), explains that the course "concentrates not so much on skills you may teach", but "on the tools that allow for decision-making that are necessary within the curatorial field", implying that "one is taught not so much about how to make an exhibition, but about why to make it, and, even more importantly, why not to" [26]. She further defines curatorial ethics as the "awareness of interrelations with which you are necessarily dealing with when you deal with curatorial practice" that requires "to train sensitivity, to look into the effects of the different doings that you chose to take on" [26]. Consequently, the major concern of curatorial ethics—as it is taught—would be the formation of an individual position that is to be identified as a *curatorial* position. In this regard, von Bismarck advocates "a very strict separation of decisions on the position from which one wants to speak from" in the art world [26]. Be it a gallerist, an artist or a curator, it is thus essential that she considers what their chosen position implies: what actions a curator settles on achieving and under which conditions these actions take place.

All in all, through this lens, the possibility of a curatorial ethic appears not in the form of some generic code of conduct for all curators to share—but rather as an individual urge to define the position, from which they want to speak; to contribute to the description of a curatorial position at large as captured by the curatorial discourse; and to develop certain techniques, through which they can deliver such contributions as an integral part of their curatorial practices. As they establish their positions and formulate their ethics, independent curators would further proceed with "judging" themselves and their colleagues. Audiences will then literally recognise their ethics in the ways

*how they talk about ethics:* how they differentiate between an ethical and unethical action; how they explain their motives and drives to curate; what personal thoughts and feelings they refer to as underlying their professional decisions and actions. As to my own experience of studying in Curating Masters programme at Goldsmiths, I had the epiphany regarding my curatorial ethic when after my presentation to the class the tutor suggested: stop trying to convince us why this subject is generally important at this time and here, and explain instead *why it is important to you*. I was reminded that as a human being I have naturally carried a massive baggage to the classroom: my upbringing, gender, race, lived experiences and biases—all of which would affect the ideas I am capable of producing, the subjects that seem important to me, and ultimately, the kind of projects that I am capable of bringing to the world as a curator. It is thus my responsibility to own up to it: interrogate it with criticality and sensitivity; understand where my reactions, drives and decisions come from; deconstruct my desire to do some projects, and stay away from others. Through this gradual process my curatorial ethic is to be configured.

If we choose to conceptualise "curatorial ethics" along those lines—as a soup of diverse individual ethics, which are likely to fight and contradict one another—what would be the practical consequences for the everyday practice of independent curators? First and foremost, we will have to let go of the habit of judging categorically their decisions and actions as either ethical or unethical. We will look at their practices with the presumption that there can be no prescriptive principles that would apply to all curators at once, which can be then used as a golden standard or converted into a code. The notion of "curatorial ethics" would rather correspond to a set of facilitative principles that do nothing more than *evaluate* what an independent curator thinks, says and does, as a demonstration of her particular way of being in the world that is disclosed by the thoughts, words, feelings and actions she is capable of at this moment. In this way, the question of curatorial ethics would be what mode of existence does this course of actions (decisions, statements, ideas, feeling) imply, as opposed to inquiring if her action (decisions, statements, ideas, feelings) is "right" or "wrong" in a given situation. In this way, curatorial ethics will be defined as "a typology of modes of existence"—to use the term by Gilles Deleuze [27] (p. 23): the collection of possible ways of *being a curator* that are available to a person today.

The "typology" or map of curatorial ethics can be further used to evaluate the practice of an individual curator (be it a self-evaluation or performed by a third party). It will look into "the qualitative difference of modes of existence" [27] (p. 23), between the potentials (how to respond to the conditions of contemporary curatorial practice? How have other curators responded?) and the individual's response. The variation of potentials here would disclose the qualitative differences amongst the way of being a curator today. For what determines the configuration of an individual curatorial ethic is not what the curator chooses to say and do; rather, it is the co-relation of the kinds of power she holds, the amount and quality of information she has about the situation, as well as the actions and responses of others, whose subject positions are continuously shaping—and being shaped by—the curator.

## 4. On Curatorial Responsibility

How does the principle of indeterminacy of curatorial practice affect the formation of other subject positions in the contemporary art field? To answer this, I will seek conceptual inspiration from Gilles Deleuze and Karen Barad—specifically relating their concepts of "becoming" (Deleuze) and "intra-action" (Barad) to the realm of curatorial practice. My decision to bring the two philosophers together deals with the similar focal point of their outlooks on ethics: the concern with the process of subject formation. Namely, both start by outlining ethics as a form of a *creative* transformational activity that is always-already engaged in the production—rather than description—of reality. In this way, both Deleuze and Barad propose the interconnection between ontology (ways of being) and ethics (ways of acting), recognising the relation of profound mutual affectivity between the two. In the context of curatorial practice, this would mean that an "underlying" ontology (i.e., all the limited ways of being, accessible to a curator today) is required to enable the creative productive power of ethics

(i.e., the number of decision-making routes that a curator can follow). At the same time, for such an ontology to be possible, a particular way of *thinking* about ethics should be in place.

Departing from this idea of interconnection, we will start by "undoing" the hegemonic dialectic dualisms that are currently used for assigning roles in the contemporary art world. In so doing, we will not aim at comprising some new roles to replace the existing ones; rather, we will deconstruct the existing ones: analysing them closely in relation to one another; and registering what are the implications of caring different attributions for those engaged in a shared art process. To give some examples, amongst the candidates for deconstruction would be contrasts such as "artist versus curator", "independent curating versus institutional curating", "intuitive versus rational", "subjective versus objective", and many other contrasts that seek to put an individual practice and its various aspect to order, by categorising. Such dualistic thinking results in the tradition of defining curatorial ethics through negation, which I am seeking to overcome: that is, through comparison between the different "kinds" of ethics that underlie the as-if-separable activities (i.e., the ethic of museum curators versus the ethic of independent curators; curatorial ethic as a code versus curatorial ethic as self-accounting; professional curatorial ethic versus the personal ethic of an individual curator; and so on). An alternative to such logic of either/or—the logic of *choosing* from the given range—the Baradian and Deleuzian outlooks on ethics enable the idea of a curator who is capable of creating the categories, by experimenting with her capacities throughout her curatorial career. In this way, the consequences of her actions will be assessed not via the dual "ethical/unethical" criteria (as stablished earlier)—but with respect to the different versions of reality, which the independent curator actualised.

The profound ethical responsibility of a curator will then be for the reality—a version of the (art) world—she creates or reinforces through her practice. For curating is never a process that an individual can undergo autonomously; it is always affected by the presence of the others with which she curates. The mechanics of this can be imagined as an enactment of "intra-action", a philosophical notion by Karen Barad [28] that helps to grasp the connectivity between a curator and the others thus denying the ontological separation between them. Barad proposes that one emerges as a subject through the encounter with the other, prior to which one would not, strictly speaking, exist or make sense as a being. This will mean replacing the image of "self-other interaction" (which is grounded in the assumption of fixed subject identities) with the assumption of a relationship of co-composition amongst those engaged in some process together. In this way, for example, the nature of relationships amongst those "sharing" a certain reality of an art world (i.e., by being engaged in the same exhibition-making project) will be seen as a continuous co-formation of subjectivities and powers. In such relationships, no one can be held as a fixed referent for identifying the other (i.e., identifying a curator as "the other" to an artist, a spectator). Rather, it should be recognised as "the mutual constitution of entangled agencies", or "intra-action" [28] (p. 33). Unlike "interaction", which assumes that there are separate individual agencies that precede the encounter, "the notion of intra-action recognises that distinct agencies do not precede, but rather emerge through [the process]" [28] (p. 33).

The same process can be understood through the Deleuzian concept of becoming. He discusses how subjects enter into relationship with one another as unfixed potentials, forming an assemblage and transforming one another through the passage of alternatives: from the multiple possibilities of what she can become—to the singular realisation, determined by the others engaged in the process. Later on, in the same book, Deleuze illustrates the concept of becoming by using the biological narrative of the orchid and the wasp in a helpful way. Evolutionary biology provides a narrative of the orchid imitating the wasp for the propagation of its species. Deleuze and Guattari correct this narrative explaining that, in order to reach their ontological determination and realise themselves as seemingly autonomous subjects ("orchid" and "wasp"), the orchid is necessarily becoming-wasp and, simultaneously, the wasp is becoming-orchid [29] (p. 12):

The orchid does not reproduce the tracing of the wasp; it forms a map with the wasp [...] What distinguishes the map from the tracing is that it is entirely oriented toward an experimentation in

contact with the real. The map does not reproduce an unconscious closed in upon itself; it constructs the unconscious.

What is essential here is that the encounter between the two entities creates a new version of reality, a new becoming. What does it mean for the orchid to become-wasp and the wasp to become-orchid? It means a simultaneous movement in two directions: *towards and with* the other. In this sense, the "other", with which one enters a relationship through such movement, destabilises the self. This relationship of becoming is not at all a way of utilising others as a means for creating and transforming the self. Rather, such a logic of becoming involves a subtle understanding of responsibility with respect to others. We do not, in Deleuze's understanding, have a responsibility to others: rather, we have a responsibility before them, facing them, in front of them. What would happen if—in the example above—we replace "wasp" with "a curator" and "orchid" with the list of other parties engaged in the contemporary arts field? The common understanding of a curator's responsibility *to* an artist (artwork, institution, audience, etc.), in which she is obligated to another by virtue of an agreement made or a duty to be had, will be undone. The responsibility will not be undertaken on behalf of others. Rather, a curator will be responsible for how she facilitates the becoming of others: by encouraging and allowing for their self-transformation; by *responding to them as she configures her own practice*, quite differently to assuming the responsibility for them as the underlying principle of her practice.

It is the profound indeterminacy of curatorial practice that fuels the transformation for all those engaged in the art process (their intra-action; or becoming with one another). It provides an individual curator with the means and drives for changing the present state of the (art) world: by doubting, questioning, and being never satisfied with it—and with herself. A curator is always becoming someone else, because being a curator is never enough. Curating gives her an obsession with the idea of a better future, as well as the energy to pursue it. The downside is that it operates by undoing (destroying) the fixed meanings, systems and structures that would have otherwise given a curator some comforting sense of control over her reactions and decisions. While the state of uncontrollable chaotic moment is likely to cause a sensation of fear and danger in a curator, the urge for responsible, responsive, ethical practice would encourage her to experiment with and challenge her very own capabilities. For, prior to experimenting, a curator will never be able to know her intellectual, emotional and physical limits that would define what kind of a practitioner she is capable of becoming. As she will be evaluating her own actions—as well as her fellow-curators—she would not ask: "What ought to be done?". Rather, she will inquire: "what *can* be done *next*, at *the very next* moment, given the quantity of power I possess, taking into account my very affinities and desires?"

It is likely that each next curator who engages in such inquiry will come up with a different answer—so that, instead of a structured discourse, they will end up with some chaotic assortment of controversial trajectories of decision-making, actions and judgement, each implying an independent curatorial ethic. Would these be of any use without artificial systematisation? Yes, in fact. The ethic of one curator will be relevant to another curator not because they are reflections of each other, but, on the contrary, because they are unprecedented; because their stories are unique and rooted in individual backgrounds and idiosyncrasies. What will then bring the curators together is not the similarity of their ethics (i.e., a code of conduct), but the exaggerated sensitivity to the movements in contemporary culture, which they all share, as captured by their controversial judgements, conflicting choices and drives to curate.

**Funding:** This research received no external funding.

**Conflicts of Interest:** The author declares no conflict of interest.

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
