# Peer review of "Curatorial Ethics and Indeterminacy of Practice"

_philosophies, doi:10.3390/philosophies5030023_

Round 1

Reviewer 1 Report

The article is an important contribution to the discussion of the curatorial ethics. The author challenges the applicability of the traditional understanding of ethical codes to the profession of a curator and suggest a novel approach to the definition of this notion. 

The author focuses on the practice of a non-affiliated curator. The clarifications as to whether the same ideas might apply to curators hired by art institutions would be welcome.

The author continuously uses the gendered pronoun, which suggests the importance of gender roles for the discussion of curatorial ethics. This might need to be further clarified or the use of the gender pronoun might need to be reconsidered.

Author Response

Many thanks for your comments. In response to them, I have added the following footnotes in the manuscript:

[1] It is the professional ethic of independent (institutionally unaffiliated, freelance) curators that is addressed in this article. Unlike institutional curators, whose role and responsibilities are defined in their contracts and who are usually “supplied” by a code of practice by an institution-employer, the practice of independent curators develops in the conditions of indeterminacy. Nothing is formally prescribed to them: to either take on board or argue and act against it. It is, therefore, the absence of any kind of formalised professional ethics for independent curators (on the one hand) and the multiplicity of controversial ethical expectations simultaneously coming from artists, institutions and other parties they encounter and/or collaborate with (on the other hand) that results in an ethically complex position I am addressing here.

[2] Throughout this article I have chosen to use “she/her” as pronouns to address an individual curator. This does not imply that the suggested perspective on curatorial ethics is not applicable to other genders, but highlights that the ideas presented are grounded in -- and determined by -- a personal, first-hand experience and practice of a female contemporary art curator.

Reviewer 2 Report

The investigation into curatorial ethics is timely and relevant, as noted by the author, there is a pressing need to evaluate ethics in relation to the role of the curator, curatorial practice and curatorial discourse. 

The study would be well served by a case study (or two or three) to elucidate the complexities of the role of the curator as described by the author, as well as where ethical responsibility falls in situations where there is censorship, controversy or closure. For example, Maria Lind's acceptance of oil money, the reception of Sam Durant's Scaffold at the Walker, the curatorial quagmire of Israeli funding in projects that include Palestinian artists such as Sao Paulo, or the artists' boycott of the Whitney Biennial. Ethics for the author remains an abstract 'quality' of curatorial practice, determined and invented by the curator based on her self-awareness of a self-identity and dependant on her proclivities and agency. Yet, Deleuze reveals ethics as a (trans)formative reality - one the author never really takes on via examples that would show the reader that curatorial ethics is complex and interconnected and difficult to determine precisely because curatorial ethics are not under the control of the curator alone. 

I support the paper's perspective, because there is a struggle taking place. A young curator wants to 'do the right thing' but is confronted with a field that is - well, 'dirty - for lack of a better word. Using the author's example, perhaps 'owning' this struggle from the outset would help. Why is she concerned with ethics? What is at stake for a young curator entering a field where there is on many ways, no defined ethical topology. 

Ethics is not simply knowing that we all arrive with baggage and have a unique perspective and background that limits and defines our ability to act. It is our obligation to the other despite our limitations, not by taking responsibility for the other (as the author rightly argues). But equally, we should not use identity politics (gender, race, nationality etc) to prescribe the limits of our ethics. I also think that the artist is conspicuously missing from this discussion and needs to be brought in for the very reason that curatorial ethics is not the sole domain of the curator.  It is telling that the author asks us to imagine the curator as the wasp -- what if the curator is the orchid?   

Author Response

Many thanks for your comments. In response to them, I have added the following footnotes:

[3] It is a professional ethic of independent (institutionally unaffiliated, freelance) curators that is addressed in this article. Unlike institutional curators, whose role and responsibilities are defined in their contracts and who are usually “supplied” by a code of practice by an institution-employer, the practice of independent curators develops in the conditions of indeterminacy. Nothing is formally prescribed to them: to either take on board, or argue and act against it. It is therefore the absence of any kind of formalised professional ethics for independent curators (on the one hand) and the multiplicity of controversial ethical expectations simultaneously coming from artists, institutions and other parties they encounter and/or collaborate with (on the other hand) that results in an ethically complex position I am addressing here. I ongoingly experience such tension in my work as an independent curator and observe it in my colleagues. The desire to reflect on this lived experience and interrogate it in-depth inspired the research leading to this article.

[4, to give examples of ethical curatorial predicaments from my practice] In 2014, I curated an exhibition for a Russian state museum and was called “unethical” by a local art critic for including “offensive contents” in a G-rated display, while neither the museum nor myself considered it inappropriate. In 2016, as a part of my PhD project, I collaborated with an artist who at the time was my research supervisor; this behaviour was recognised as “unethical” by the University Ethics Committee. From 2018 when I started Exposed Arts Projects (an art-research hub and exhibition venue in London) I have been repeatedly called “unethical curator” by artists and collaborators who do not share the values and non-judgemental policy of my organisation.